# The Effect of Porosity on the Thermal Conductivity of Highly Thermally Conductive Adhesives for Advanced Semiconductor Packages

**DOI:** 10.3390/polym15143083

**Published:** 2023-07-18

**Authors:** Hyun-Seok Choi, Jeong-Hyun Park, Jong-Hee Lee

**Affiliations:** Advanced Materials Research Center, SMT Corporation, Suwon-si 16643, Gyeonggi-do, Republic of Korea

**Keywords:** thermal conductivity, thermally conductive adhesive, aluminum nitride, aluminum oxide, Bruggeman asymmetric model

## Abstract

This study suggests promising candidates as highly thermally conductive adhesives for advanced semiconductor packaging processes such as flip chip ball grid array (fcBGA), flip chip chip scale package (fcCSP), and package on package (PoP). To achieve an extremely high thermal conductivity (TC) of thermally conductive adhesives of around 10 Wm^−1^K^−1^, several technical methods have been tried. However, there are few ways to achieve such a high TC value except by using spherical aluminum nitride (AlN) and 99.99% purified aluminum oxide (Al_2_O_3_) fillers. Herein, by adapting highly sophisticated blending and dispersion techniques with spherical AlN fillers, the highest TC of 9.83 Wm^−1^K^−1^ was achieved. However, there were big differences between theoretically calculated TCs that were based on the conventional Bruggeman asymmetric model and experimentally measured TCs due to the presence of voids or pores in the composites. To narrow the gaps between these two TC values, this study also suggests a new experimental model that contains the porosity effect on the effective TC of composites in high filler loading ranges over 80 vol%, which modifies the conventional Bruggeman asymmetric model.

## 1. Introduction

To date, various types of highly thermally conductive materials have been researched based on filler types, shapes, and content [1,2,3,4]. According to the theoretical models that have already been set up to calculate the TCs of organic–inorganic composites, many researchers have attempted to predict effective TCs by comparing the measured TCs with the calculated values from these models. However, the main limitation of these studies is that they were generally conducted in a low filler loading range of 10–30 vol% [5,6,7]. Practically, there have been few studies related to highly thermally conductive materials that are in a high filler loading range of over 70 vol% and have TCs of over 5 Wm^−1^K^−1^ in the technical aspects. The main reason for the large gaps between the values calculated from theoretical models and those experimentally measured is that the effect of micro-voids between inorganic fillers and polymer matrixes during dispersion and solidifying processes has not been studied.

Therefore, this study was conducted to determine practically highly thermally conductive materials containing spherical ceramic fillers in the high filler loading ranges, especially over 80 vol%, from a technical perspective, as well as to validate the causes of the differences in TCs between theoretical and experimental values. In particular, this study focused on the main factors that affect decreasing measured TCs compared with theoretically calculated ones from the perspective of the effect of micro-voids that exist or generate at the interfaces between inorganic ceramic fillers and organic matrix polymers during processing. These micro-voids are naturally generated from the bad wettability of hydrophobic matrix polymers against hydrophilic ceramic fillers [8] and then increase the thermal resistance between inorganic fillers and matrix polymers.

To achieve high TCs in composite materials, there are two technical approaches. One method involves increasing the TCs of matrix polymers, which is well known to be more effective than increasing the TCs of inorganic fillers [9]. However, increasing the TCs of polymers is not easy and is closely related to the original properties of polymer raw materials. The other method involves using highly thermally conductive filler materials or increasing the filler volume ratio, which is a relatively easy process that is capable of being conducted in both academic and industry settings.

There are three main technical trends to achieve high TCs in composite materials using highly thermally conductive inorganic fillers. The first involves using metallic fillers for applications where there are no electrical insulation issues. Highly thermally conductive fillers such as aluminum (200 Wm^−1^K^−1^), copper (400 Wm^−1^K^−1^), and silver (400 Wm^−1^K^−1^) are typically used as fillers in thermally and electrically conductive adhesives for semiconductor packaging applications in which electrical insulation is not needed in general [10,11,12].

The second involves using hybrid fillers to decrease the interfacial thermal resistance by coating two-dimensional materials such as carbon nanotubes (CNTs), graphene, and boron nitride on the surface of spherical ceramic fillers. By attaching two-dimensional materials, it is expected that contact areas among inorganic fillers will increase from point-to-point contact to plane-to-plane contact [13,14,15,16].

The third involves using highly thermally conductive ceramic fillers such as aluminum nitride (AlN; 180 Wm^−1^K^−1^), boron nitride (BN; 110 Wm^−1^K^−1^), silicon carbide (SiC; 140 Wm^−1^K^−1^), magnesium oxide (MgO; 120 Wm^−1^K^−1^), and high-purity aluminum oxide (99.99% Al_2_O_3_; 40 Wm^−1^K^−1^). This method is currently being widely researched and is expected to be the most practical from an industrial perspective. Since spherization technology for irregularly shaped AlN or flake-shaped BN powders was recently developed and commercialized, it is now possible to use large spherical AlN or BN particles, whose diameters are dozens of micrometers and are widely studied, to develop extremely thermally conductive materials over 10 Wm^−1^K^−1^ [17,18]. Usually, these types of spherical AlN or BN powders are made via the agglomeration process using processes such as the spray drying technique and sintering [19]. In some extreme applications, synthetic diamond fillers are also used as thermally conductive fillers, despite their high price [20,21]. 

At present, using spherical AlN or BN powders is considered to be the best and only way to achieve high TC and electrical insulation simultaneously. However, such nitrated materials as AlN or BN easily react with moisture in the air, and their surface properties are chemically changed from the highly thermally conductive nitride (-N) phase to the low thermally conductive hydroxide (-OH) phase. Moreover, the ammonia gases generated by the chemical reaction act as a curing inhibitor for polydimethylsiloxane or epoxy polymers during the heat curing process [22].
AlN + 3H_2_O ⇒ Al(OH)_3_ + NH_3_ (↑)
BN + 3H_2_O ⇒ B(OH)_3_ + NH_3_ (↑)

Therefore, it is mandatory to protect the surfaces of AlN or BN fillers by coating them with silane compounds or organic metal complexes. Unsatisfactorily, these surface coatings also act as thermal transfer barriers because organic materials generally have low TCs compared with inorganic materials such as ceramics and metals. In spite of the demerits of these materials, this method is accepted as the most practical way to achieve highly thermally conductive composite materials. Table 1 shows the three kinds of methods for accomplishing high TC, as well as the merits and demerits of each one. 

Another important aspect of researching and designing compositions for thermally conductive composites is predicting the TCs of hybrid materials. There have been many numerical models trying to calculate accurate TCs of organic and inorganic composites [23,24,25,26]. However, it is too difficult to predict the exact TC values of composite materials in high filler loading ranges. In this study, the simplified Bruggeman asymmetric model was applied because it is known to be more meaningful for spherical and relatively high filler loading systems compared with other numerical models. The Bruggeman asymmetric model was advanced from the Bruggeman symmetric model (BSM), which has been widely used to predict the electric conductivity of a composite and has also been applied for TC prediction in some cases. However, BSM has difficulty predicting the exact TC of a composite system when there is a big difference in TCs between inorganic fillers and matrix polymers [27]. On the other hand, the Bruggeman asymmetric model has been developed to be applied to systems where there is a big difference in the TCs of the component materials. Moreover, BSM is just adequate for filler loading ranges up to the percolation threshold, but the Bruggeman asymmetric model is applicable to higher filler loading ranges. 

## 2. Materials and Methods

### 2.1. Preparation of the Materials

In this study, five kinds of inorganic materials were used as fillers, while polydimethylsiloxane (PDMS) polymers were used as matrix binders. A general grade of aluminum oxide filler (DAW series, Denka Company Limited, Japan; purity of 99.8%), a high purity grade of aluminum oxide filler (AA series, Sumitomo Chemical Co., Ltd., Japan; purity of 99.99%), a 0.3 wt% graphene-coated aluminum oxide filler (graphene; xGnP^®^, grade H, XG Science, USA; aluminum oxide; AA series), a spherodized aluminum nitride filler (FAN-f series, Furukawa Denshi, Japan), and a silver-coated copper filler (Nopion Corporation, Republic of Korea) of three different sizes were used. In the case of the Al_2_O_3_-based fillers, sizes of 20, 3, and 0.25 µm were used, based on the ratio of Horsfield’s close-packing rule. In the case of the AlN fillers, 30, 5, and 0.5 µm were used to match the ratio of the close-packing rule. Lastly, in the case of the Ag-coated Cu fillers, 20 µm was used solely as the reference for metal-based thermal interface materials. 

The fillers were dispersed homogeneously in a blended polymer of a vinyl-terminated polydimethylsiloxane (PDMS) polymer (VEP-100, KCC Corporation, Republic of Korea; viscosity of 100 mPa·s) and a hydrogen-terminated PDMS polymer (MHBP-011, KCC Corporation, Republic of Korea; viscosity of 10 mPa∙s) at a ratio of 10:1. The VEP-100 and MHBP-011 polymers used in this study have extensively low volatile matters under 10 ppm for the purpose of minimizing the effect of pore generation in composites during the compounding and curing processes due to the polymers’ own volatile matters. The compositions were designed for each filler system, as shown in Table 2. The dispersion process was operated by a planetary centrifugal mixer (ARV-310, manufactured by Thinky Corporation, Japan). The dispersing process was conducted at a revolving speed of 2000 rpm for 3 min for all three kinds of fillers, respectively. The operating order of the dispersing fillers was from large to small in sequence. Then, they were processed to manufacture sheets of 3 mm thickness by a typical hot press to measure the TC and porosity. The pressing pressure was 102 kgf·cm^−2^, and the curing condition of the compound was 150 °C for 30 min. 

### 2.2. Characterization

The TC, thermal resistance, and bond line thickness (BLT) of the uncured and cured specimens were measured by a heat flow method following the ASTM D5470 standard (T3Ster DynTIM S, Mentor, USA). The TC was derived from the slope of thermal resistance versus the BLT curve, and the TC of the specimen was the reciprocal of the slope. The thermal resistance values in Table 3 were checked at the BLT. The viscosity of the compounds before heat curing was checked by a rheometer (MCR-302, Anton Paar, Austria). The viscosity was decided as the value checked by a spindle of PP-25 (plate-to-plate type) at a shear rate of 10 s^−1^. The density of the cured specimens was measured by an electronic densimeter (GR-200, AND Co., Ltd., Japan) following the ASTM D792 standard. The volume resistivity was measured by a high-resistivity meter (Hirestar-UX, Mitsubishi Chemical Analysistech Co., Ltd., Japan). Finally, a scanning electron microscope (SEM) (SNE-4500M, SEC Co., Ltd., Republic of Korea) was used to observe the shape and microstructure of the filler materials. Five specimens for each composition were made and measured four times for each specimen. The average values were used for characterization.

## 3. Results and Discussion

### 3.1. Typical Properties of Highly Thermally Conductive Composites

Each composition made of inorganic fillers and PDMS polymers was characterized in terms of its physical and thermal properties, as shown in Table 3. The viscosity of the uncured compounds varied with filler content and shape. The compound using high-purity Al_2_O_3_ fillers showed lower viscosity than the compound using general-grade Al_2_O_3_ fillers by nearly 15% at the same level of filler content. Comparing the Al_2_O_3_ filler system with the AlN filler system, the AlN filler system showed nearly double the viscosity, despite its filler content being almost at the same level. The lower viscosity of the high-purity Al_2_O_3_ filler system compared with the other systems was considered to have originated from the surface morphology of the fillers. Due to the fact that the high-purity Al_2_O_3_ fillers were made by the CVD (chemical vapor deposition) process, their surfaces were smooth and clean. Additionally, the low surface roughness and non-existence of functional groups such as hydroxyl (-OH) and carboxyl (-COOH) are particularly advantageous during wetting and dispersing processes in PDMS polymers [28]. The morphology of the high-purity and general-grade Al_2_O_3_, graphene-coated Al_2_O_3_, and AlN fillers is shown in Figure 1. Because the general-grade Al_2_O_3_ and AlN fillers were made by a kind of thermal spray method, the shape was completely spherical; however, the surface morphology was not even, and there were some voids on the surface, as shown in Figure 1a,b. When the surfaces of the high-purity Al_2_O_3_ fillers were coated with graphene materials, the compound viscosity of the graphene-coated Al_2_O_3_ filler system increased much more than that of the non-coated filler system. However, the effectiveness of the graphene coating in enhancing the TC of the composite was enormous. The TC increased by over 50%, from 3.83 to 6.17 Wm^−1^K^−1^, due to the graphene coating, in spite of the lower filler content of the graphene-coated Al_2_O_3_ filler system. This effect was considered to be attributed to the increasing contact area between two fillers, from point-to-point contact to area-to-area contact, as expected [29]. Another issue was that the electric insulation property was also affected by the graphene coating because the graphene material itself is a naturally good electrical conductor, as can be seen in Table 3. Therefore, in the case of using carbon-based materials for coating to make core–shell-type fillers, a decrease in volume resistivity should also be considered. 

As for the TCs of the inorganic filler composites, the most successful way to accomplish high TCs was via the use of AlN fillers (9.83 Wm^−1^K^−1^), followed by using high-purity fillers (7.8 Wm^−1^K^−1^). However, considering the heat transferring performance, using high-purity Al_2_O_3_ fillers was potentially the most effective way to dissipate the heat in semiconductor packaging applications because, at the same applied conditions, the high-purity Al_2_O_3_ filler system had the lowest thermal resistance compared with the AlN filler system. Additionally, because the high-purity Al_2_O_3_ filler system showed the lowest viscosity value, the possibility of containing voids or forming pores during the compounding and solidifying processes would be the lowest. 

### 3.2. Analysis of the Differences between the Theoretically Calculated and Experimentally Measured TC Values

Using the simplified Bruggeman’s asymmetric model, the differences between the TCs calculated using the theoretical model and those actually measured by instruments were analyzed. The results are shown in Table 4. In the case of using the high-purity Al_2_O_3_ filler system, the gap between the theoretical and experimental TC values was the lowest. In the case of using the AlN filler system, the highest TC was achieved, but the gap was also larger. It should be noted that in the case of using mono-sized Ag-coated Cu fillers, the gap was the largest. Based on these results, the most influential factor for achieving a theoretical TC value for inorganic fillers was the close-packing structure of the fillers. The size of the commercialized AlN fillers did not vary compared with the commercialized Al_2_O_3_ fillers because the technology related to manufacturing spherical AlN fillers is newly developed and manufacturers are also limited.

It was considered that the rough surfaces of the AlN and general-grade Al_2_O_3_ fillers (as shown in Figure 1) had a bad effect on the wettability of the filler surfaces with the PDMS polymers. However, in order to confirm this phenomenon, extremely sophisticated analysis techniques are needed to observe the interfaces between inorganic fillers and matrix polymers. Therefore, an alternative technique was tried to analyze the indirect effect of the interface voids on the differences in the TCs of various composites. By measuring the densities of the cured specimens and comparing them with the theoretical density of the designed compositions, the porosity could be calculated, and it was reasonable to analyze the relationship between the porosities and TCs of each composite.

Figure 2 shows the gaps of the TCs between the theoretical and experimental values in accordance with the filler system. The gaps of the AlN and graphene-coated Al_2_O_3_ filler systems were larger than those of the other filler systems. These big discrepancies were considered to be due to the rough surfaces of the two filler systems, as mentioned before. More meaningfully, the gaps between the TCs diverged as the filler content increased in all filler systems. When the filler loading content exceeded 80 vol%, the theoretically and experimentally measured TCs became completely different. Therefore, in extremely high filler loading ranges of 80~90 vol%, it is necessary to adjust the theoretical models to reflect the porosity effect. In the simplified equation of the Bruggeman asymmetric model, the volume fraction Φ needs to be modified to reflect the porosity effect. 

### 3.3. Suggestion of a Modified Theoretical Model to Predict Practical Thermal Conductivity

A modified theoretical model to predict more practical TCs of composites that contain an extremely high filler content was suggested based on the simplified Bruggeman asymmetric model [27]. This model was specially designed for systems composed of spherical inorganic fillers and matrix polymers and for cases of a high filler content over 60 vol%. Due to the Bruggeman symmetric model (BSM) and the original equation for the Bruggeman asymmetric model being based on the percolation theory, these models are known to be beneficial for predicting high thermal conductivity, usually 3~5 Wm^−1^K^−1^ in general. The simplified Bruggeman asymmetric model is as follows: (1)$$1-\Phi =\frac{k_{e}-k_{d}}{k_{c}-k_{d}}{\left(\frac{k_{c}}{k_{e}}\right)}^{1/3}$$
where *Φ* is the volume fraction of the fillers, *k_e_* is the effective TC of a composite, *k_d_* is the TC of the fillers, and *k_c_* is the TC of a matrix polymer. Equation (1) was applied to this study to deal with high filler loading situations and systems that contain nearly spherical Al_2_O_3_ and AlN particles. In the Bruggeman asymmetric model, the effective TC of a composite can be easily calculated by inserting parameters of *Φ*, *k_d_*, and *k_c_*. *Φ* can be calculated by the relationship between vol% and density of component materials. *k_d_* and *k_c_* are the well-known material constants in literature [30]. In this study, *k_c_* was considered to be 0.16 Wm^−1^K^−1^ as the silicone polymer was used as a matrix polymer.

To evaluate the porosity effect on the TC measurement more accurately and quantitatively, it was necessary to select the system with the minimum noise interference. From this point of view, the high-purity Al_2_O_3_ filler system was selected because the compound viscosity was relatively low compared with the other filler systems and it was possible to eliminate the noises generated during specimen preparation and TC measurement. Table 5 shows the porosity values with high-purity Al_2_O_3_ filler systems of varying filler contents. As can be seen in Table 5, the porosity increased as the filler content increased due to deterioration of wettability and dispersibility. In order to adjust the theoretical TCs to reflect the porosity effect, a correction factor should be inserted into the simplified Bruggeman equation. Because porosity is a factor that is closely related to the filler volume fraction, it was reasonable to insert a correction factor into the left side of Equation (2) as follows:(2)$$1-\Phi^{\left(1+p\right)}=\frac{k_{e}-k_{d}}{k_{c}-k_{d}}{\left(\frac{k_{c}}{k_{e}}\right)}^{1/3}$$
The porosity of *p* in Equation (2) was derived from the differences between the theoretical and measured densities. In all composite systems, it is not avoidable to contain defects like voids or pores during processing. And because these defects usually exist at the interfaces between inorganic fillers and matrix polymers, the porosity does not only affect the filler volume fraction but also acts as thermal transfer barriers. In the proposed model, the porosity increase eventually induces a decrease in the effective TCs of the composites. However, there were still large differences in the higher filler loading ranges, requiring compensation of the term related to the TCs of the fillers on the right side of Equation (2), as shown in Figure 3.

Additionally, it is rare to find studies focused on the porosity effect on the TC of thermally conductive fillers and composites. To achieve this, it is necessary to employ a critically precise microstructure analysis of interfacial voids by cross-sectional SEM and TEM in future works.

## 4. Conclusions

To achieve highly thermally conductive adhesives, three kinds of filler materials were tried. Regarding thermal conductivity itself, the highest thermal conductivity of 9.83 Wm^−1^K^−1^ was achieved using spherical AlN fillers. However, in terms of heat transfer performance, considering thermal resistance and bond line thickness together, using high-purity Al_2_O_3_ fillers of 99.99% was considered to be the most effective way to transfer heat. The thermal resistance of the high-purity Al_2_O_3_ filler system was 0.52 KW^−1^, which is much lower than that of the spherical AlN filler system, 0.85 KW^−1^, in spite of the lower filler content. This is mainly due to the fact that the smooth and clean surface of high-purity Al_2_O_3_ fillers minimizes the possibility of containing interfacial voids during compounding and solidifying processes. These two compositions can be promising candidates for thermally conductive adhesives in advanced semiconductor packaging applications.

In order to adjust the gaps between the theoretically and experimentally measured thermal conductivity values, the porosity effect should be considered in high filler loading ranges. A modified Bruggeman asymmetric model including a correlation factor to the term of the filler volume fraction was suggested, and it was meaningful to reduce the gaps between the theoretical and experimental thermal conductivities in high filler loading ranges over 80 vol% in particular. However, there is still a requirement to narrow the gap between the theoretical and experimental thermal conductivities of composite materials, and it is necessary to analyze the interfacial voids more quantitatively. To achieve a more precise model, microstructural analyses by high-resolution SEM and atomic force microscope will be conducted for the purpose of validating the porosity effect in future works. By applying a more accurate correlation factor, the modified Bruggeman asymmetric model can be more practical and can then be used as composition guidelines for designing and manufacturing thermal interface materials.

## Figures and Tables

**Figure 1 polymers-15-03083-f001:**
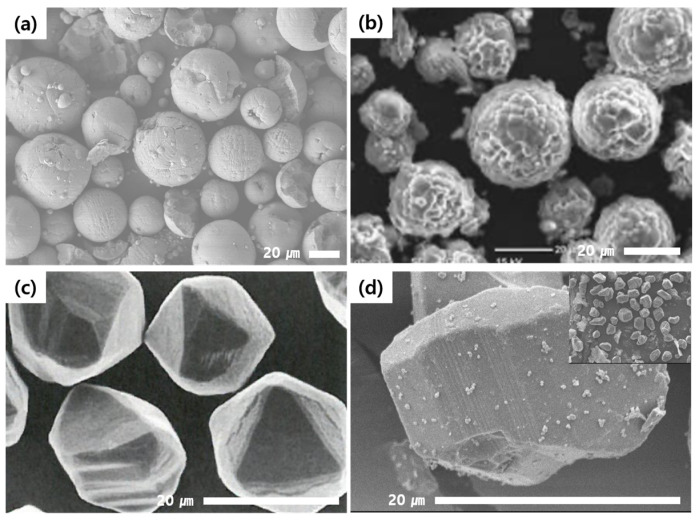
Scanning electron microscope (SEM) images of (**a**) general-grade Al_2_O_3_ fillers, (**b**) AlN fillers, (**c**) high-purity Al_2_O_3_ fillers, and (**d**) graphene-coated high-purity Al_2_O_3_ fillers.

**Figure 2 polymers-15-03083-f002:**
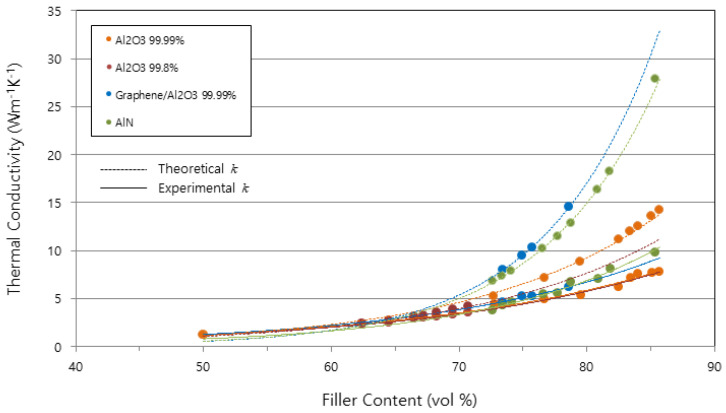
The gaps in thermal conductivity between the theoretically and experimentally measured values.

**Figure 3 polymers-15-03083-f003:**
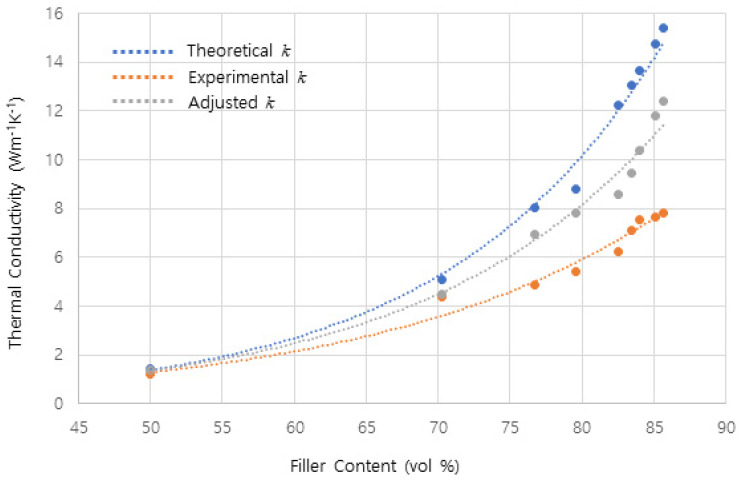
The adjusted theoretical thermal conductivity reflecting the porosity effect of the interfacial voids in the spherical AlN filled composites.

**Table 1 polymers-15-03083-t001:** The filler concepts for accomplishing high thermal conductivity.

Technologies	Metallic Fillers	Hybrid Fillers	High *k* Ceramic Fillers
Structure	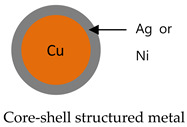 - Ag coated Cu - Ni coated Cu	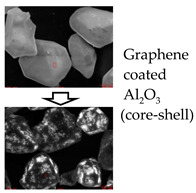	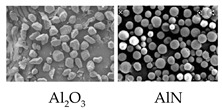 - High Purity Al_2_O_3_ (99.99%) - AlN, SiC, MgO
Merit	- High thermal conductivity	- Cost	- Electrical insulation - Cost
Demerit	- Cost (∵high density) - Electrically conductive	- Difficult to load - Electrically conductive	- Need surface treatment

**Table 2 polymers-15-03083-t002:** Experimental compositions composed of inorganic fillers and polydimethylsiloxane (PDMS) polymers.

Sample		Composition (wt%)					Filler wt%	Filler vol%	Remark
		20 µm	3 µm	0.25 µm	PDMS	Total			
Al_2_O_3_-based	AO1	56.0	18.0	21.3	4.7	100	95.3	83.4	High purity grade
	AO2	56.0	18.0	21.3	4.7	100	95.3	83.4	General grade
	AO3	54.0	17.0	21.0	6.4	100	93.6	78.6	Graphene -coated
AlN-based	AN1	54.0 (30 µm)	17.0 (5 µm)	24.1 (0.5 µm)	4.9	100	95.1	85.4	
Ag-coated Cu	AC1	92	-	-	8	100	92.0	54.9	

**Table 3 polymers-15-03083-t003:** Physical and thermal properties of inorganic fillers and PDMS compounds.

Sample	Filler Composition (wt%)			Filler Content (wt%)	Filler Content (vol%)	η (Pa·s)	K (Wm^−1^K^−1^)	R_th_ (KW^−1^)	BLT (µm)	ρ (Ω•cm)	Remark
	20 µm	3 µm	0.25 µm								
AO1	56.0	18.0	21.3	95.3	83.4	70	7.12	0.52	30	1.6 × 10^14^	Al_2_O_3_ 99.99%
AO2	56.0	18.0	21.3	95.3	83.4	80	3.83	1.84	56	1.2 × 10^14^	Al_2_O_3_ 99.8%
AO3	54.0	17.0	21.0	93.6	78.6	350	6.17	0.86	27	3.7 × 10^10^	graphene/Al_2_O_3_ 99.99%
AN1	56.0 (30 µm)	17.6 (5 µm)	20.5 (1 µm)	95.1	85.4	150	9.83	0.85	47	2.2 × 10^13^	AlN 99.5%
AC1	92	-	-	92.0	54.9	130	4.14	0.78	20	4.6 × 10^4^	Ag/Cu

**Table 4 polymers-15-03083-t004:** Differences between the theoretically and experimentally measured thermal conductivities.

Samples	AO1	AO2	AO3	AN1	AC1
Theoretically calculated value	11.92	10.20	14.55	27.80	18.68
Experimentally measured value	7.80	3.83	6.17	9.83	4.14
Gap	4.12 (34.6%)	6.37 (62.5%)	8.38 (57.6%)	17.97 (64.6%)	14.54 (77.8%)
Estimated reason	-Interfacial micro-void -Incomplete Packing	-Purity -Interfacial micro-void -Incomplete packing	-Rough surface of filler -Interfacial micro-void -Incomplete packing	-Sphericity -Interfacial micro-void -Incomplete packing	-Small size -Interfacial micro-void -Unclose packing
Remark	k_Al2O3_ = 39	k_Al2O3_ = 30	k_Al2O3_ = 39, k_Graphene_ = 390	k_AlN_ = 150	k_Ag_ = 405, k_Cu_ = 400

**Table 5 polymers-15-03083-t005:** The effect of porosity on the thermal conductivities and adjusted values by the modified Bruggeman asymmetric model.

Sample	AO-1	AO-2	AO-3	AO-4	AO-5	AO-6	AO-7	AO-8	AO-9
wt%	80.12	90.50	93.00	94.00	95.00	95.30	95.50	95.83	96.02
vol%	50.00	70.27	76.72	79.54	82.50	83.42	84.04	85.08	85.68
Theoretical ρ (g/cm^3^)	2.4649	3.0670	3.2587	3.3423	3.4302	3.4575	3.4760	3.5068	3.5248
Measured ρ (g/cm^3^)	2.4546	3.0412	3.2163	3.2915	3.3671	3.3922	3.3877	3.4195	3.4275
Porosity *	0.0054	0.0084	0.0130	0.0152	0.0184	0.0189	0.0254	0.0249	0.0276
Theoretical k (Wm^−1^K^−1^)	1.45	5.08	8.05	8.79	12.25	13.07	13.68	14.73	15.39
Measured k (Wm^−1^K^−1^)	1.22	4.40	4.88	5.40	6.23	7.12	7.53	7.66	7.80
Adjusted k (Wm^−1^K^−1^)	1.39	4.50	6.95	7.80	8.60	9.45	10.40	11.80	12.41

* Porosity = (theoretical density − measured density)/(theoretical density).

## Data Availability

Not applicable.

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
