# Peer review of "The Effect of Porosity on the Thermal Conductivity of Highly Thermally Conductive Adhesives for Advanced Semiconductor Packages"

_polymers, 2023, doi:10.3390/polym15143083_

Round 1

Reviewer 1 Report

The paper entitled "The Effect of Porosity on the Thermal Conductivity of Highly Thermally Conductive Adhesives for Advanced Semiconductor Packages" demonstrates valuable insights into the development of thermally conductive adhesives for semiconductor packaging. There are some minor and major comments below to help the readers to be more beneficial from the Paper.

1.      The abstract is written general. What the specific objective of the study is. The abstract could benefit from providing a concise statement on the novelty or contribution of the new model proposed in the study.

  1. Introduction, line 61-63 which percentage of CNT and GNP would be appropriate for conductivity. Refer to the references blow for more information:

[a] Sensitive response of GNP/epoxy coatings as strain sensors: Analysis of tensile-compressive and reversible cyclic behavior. Smart Materials and Structures, 2019, 29(6), 065012.

[b] Impedance analysis for condition monitoring of single lap CNT-epoxy adhesive joint. International Journal of Adhesion and Adhesives2019, 88, 59-65.

3.      How many specimens were created, and how has the reproducibility of the results been confirmed?

4.      The authors examined the impact of porosity on thermal conductivity and developed a model. Can the authors offer an equation that encompasses all relevant parameters to describe this relationship accurately?

5.      The discussion on the modified Bruggeman asymmetric model is insightful, but it would be beneficial to provide a brief explanation of how this model is modified and how the correlation factor affects the results.

6.      The conclusion could be strengthened by suggesting potential ways for future research or highlighting specific challenges that need to be addressed to improve the theoretical-experimental gap in thermal conductivities and analyze interfacial voids more quantitatively. It would be valuable to emphasize the practical significance of the research and how it advances the field of semiconductor packaging. Use bullets to emphasise the main achievements of the paper

Author Response

Dear Sir.

Thank you.

Reviewer 2 Report

I enjoy reading the paper. However, the authors need to address the following comments before the acceptance.

1. More reference should be added to line 54 – 60 as a lot of information cannot be tracked.

2. The sequence for presenting three methods in Table 1 should match the sequence how the authors described them in the manuscript.

3. In line 173, the authors emphasized the importance role played by the surface roughness on TC. Therefore, I suggest authors using AFM to get the morphology information, which could help quantify the roughness.

4. Details about how to calculate theoretical values in Table 4 need to be presented in the main manuscript.

5. The authors should be pay attention to the results in Figure 2, which showed TC values are almost the same for different filler under 65% filler content. Does it make sense?

Minor revision.

Author Response

Dear Sir.

Thank you.

Reviewer 3 Report

The manuscript is devoted to thermal conductivity of adhesives used in advanced microelctronic packaging. The overall impression from reading is positive. The manuscript is well structured. Authors provide detailed introduction on materials and methods used to date, and on the existing problems. Theoretical and experimental parts are present. The results are clearly described. The work may be interesting for many fields dealing with thermally conductive adhesives, not limited to semiconductor industry. It can be published with minor changes. I recommend only the formal corrections:

1. The authors state that the proposed model for thermal conductivity is new. However, other models are not reviewed. Only references are provided, but no details are given. It is hard to understand, how the proposed model differs from those reported previously, and what is the novelty. I recommend to review the existing models in Introduction and to clarify the new approach.

2. The manuscript needs proofreading. Some typos are present, like “filer” at line 170.

Author Response

Dear Sir.

Thank you.

Round 2

Reviewer 1 Report

I have carefully reviewed the second version of the manuscript mentioned above and I am pleased to inform you that I am in favor of accepting it for publication.